# Targeting Cleavage of C-Terminal Fragment of Cytoskeletal Filamin A in Cancers

**DOI:** 10.3390/cells13161394

**Published:** 2024-08-21

**Authors:** Ozgur Cakici, Sashidar Bandaru, Grace Yankun Lee, Dyar Mustafa, Levent M. Akyürek

**Affiliations:** 1Sabri Ülker Center for Metabolic Research, Department of Molecular Metabolism, Harvard T.H. Chan School of Public Health, Boston, MA 02115, USA; ocakici@hsph.harvard.edu (O.C.); gylee@hsph.harvard.edu (G.Y.L.); 2Department of Clinical Pathology, Sahlgrenska University Hospital, Västra Götalandsregionen, 413 45 Gothenburg, Sweden; sashidar.bandaru@medkem.gu.se; 3Department of Laboratory Medicine, Institute of Biomedicine, University of Gothenburg, Sahlgrenska Academy, 405 30 Gothenburg, Sweden; dyar.mustafa@gu.se

**Keywords:** cytoskeleton, filamin, cleavage, cancer, inhibition

## Abstract

Human cancers express altered levels of actin-binding cytoskeletal filamin A (FLNA) protein. FLNA in mammals consists of an actin-binding domain at its N-terminus that is followed by 24 immunoglobulin-like repeat modules interrupted by two hinge regions between repeats 15–16 and 23–24. Cleavage of these hinge regions produces a naturally occurring C-terminal 90 kDa fragment of FLNA (FLNA^CT^) that physically interacts with multiple proteins with diverse functions. This cleavage leads to actin cytoskeleton remodeling, which in turn contributes to cellular signaling, nucleocytoplasmic shuttling of transcriptional factors and nuclear receptors, and regulation of their transcriptional activities that are important for initiation and progression of cancers. Therefore, recent studies have proposed blocking FLNA cleavage as a means of cancer therapy. Here, we update how FLNA cleavage has been targeted by different approaches and their potential implications for future treatment of human cancers.

## 1. Structure and Function of Filamin A

The family of actin-binding cytoskeletal filamins consists of three members, filamin A (FLNA), FLNB, and FLNC [1]. Filamins exist in both homo- and heterodimeric structures. Both FLNA and FLNB are structurally similar to each other and abundantly expressed throughout the body, whereas expression of FLNC is limited to cardiac and skeletal muscles. Structurally, filamins have an actin-binding domain (ABD) at their N-terminus, followed by a rod-domain and a dimerization domain at the carboxy-terminus [2]. The ABD has two CH domains for actin-binding. FLNA exists as V-shaped dimers that can link F-actin into parallel bundles or orthogonal networks (Figure 1). The flexibility of FLNA and the actin–FLNA ratio determines the type of actin structure that forms. Additionally, transmembrane receptors can be linked to the actin cytoskeleton by FLNA.

In addition to its widely known cytoskeletal function, FLNA has surprisingly been shown to interact with a functionally diverse group of molecules involved in the regulation of cellular signaling as a scaffold and transcriptional activity [3,4]. The number of known interacting partners of FLNA is increasing, the majority of which interact with C-terminal fragments of FLNA (FLNA^CT^) produced by cleavage of its hinge 1 (H1) domain by calpains (Figure 2). FLNA^CT^ does not only bind to these molecules, but also mediates nuclear transport and even transcriptional activity by being a part of the transcriptional complex within the nucleus [3]. Initial detection of FLNA within the nucleus has been considered as nuclear contamination of this abundantly expressed cytoskeletal protein. However, these discoveries on the nuclear presence of FLNA later revolutionized the classical concept of cytoskeletal function.

In this field, human melanoma cells that are deficient for FLNA (M2 cells) as compared to M2 cells stably expressing FLNA (A7 cells) have been extensively used [5]. Impaired filopodia formation and cellular migration in FLNA deficiency has been reported [6]. Since then, these melanoma cell lines have been used to study the function of FLNA, both in vitro and in vivo, and identify interaction partners in health and disease.

As the structure of the actin cytoskeleton is finely organized partly by actin-binding FLNA during cellular migration, cells deficient in FLNA fail to reach the right places during embryogenesis, thereby causing malformations. In mice, intrauterine deficiency of Flna is lethal, dominantly due to severe cardiovascular malformations [7]. In humans, mutations in the *FLNA* gene are associated with devastating malformations affecting cardiovascular organs, craniofacial structures, the skeleton, brain, viscera, and urogenital tract [8].

Nuclear localization of FLNA surprised the scientific community as cellular structural proteins have been recognized as cytoplasmic proteins and their nuclear presence have been considered as contamination for a long time. However, the presence of actin and actin-binding proteins in the cell nucleus is no longer controversial at all. Currently, cytoskeletal proteins detected in the nucleus are known to be involved in chromatin remodeling, transcription, and nuclear transport [9]. Several of these proteins have been implicated in chromosome movement. Interestingly, once cytoskeletal FLNA is a nucleolar protein; it suppresses ribosomal RNA gene transcription [10]. Recent literature data indicate that an increase in the level of many actin-binding proteins including FLNA is involved in the reorganization of the actin cytoskeleton, which is associated with the induction of the epithelial–mesenchymal transition process, metastasis, and a worse prognosis for cancer patients [11].

## 2. Interaction Partners of Filamin A Involved in Cancers

Both cytoplasmic and nuclear FLNA are involved in cancer-related phenotypes depending upon the type and stage of cancer, where cytoplasmic FLNA is particularly involved in cancer cell migration and proliferation, while nuclear FLNA is involved in cancer cell signaling and transcription. In melanoma cell lines, transcriptional factor HIF-1a interacts with FLNA^CT^ and then is translocated into the nucleus to bind to the VEGF promoter to increase xenograft tumor progression and angiogenesis [12] (Table 1). Increased expression of FLNA is associated with high-risk neuroblastoma tumors in patients, and inhibition of increased expression of FLNA by shRNA reduces xenograft neuroblastoma growth in mice [13]. In high-risk neuroblastoma cell lines, transcriptional factor STAT3 interacts with FLNA, promoting tumor cell growth and phosphorylation and signaling [13]. In multiple tumor cell lines that are induced by HGF/c-MET signaling through AKT, FLNA regulates oncogenic responses by binding to SMAD2 [14]. Particularly in cancer-associated fibroblasts, the transcriptional factor androgen receptor (AR) binds to FLNA^CT^, mediates nuclear translocation of AR, and subsequently increases incidence of prostate cancer [15]. In breast cancer cells, FLNA interacts with tumor suppressor gene BRCA1 and stabilizes the early stages of DNA repair, and this interaction acts as checkpoint in tumor progression [16]. BRCA2 also interacts with FLNA^CT^ to connect cytoskeletal signal transduction to DNA damage response pathways [17], suggesting a functional role of FLNA in the recovery from G_2_ arrest and subsequent mitotic cell death after DNA damage [18]. Thus, cytoskeletal signal transduction mediated by FLNA may also be targeted for DNA damage-based cancer therapy [19]. FLNA interacts with GPCR and negatively downregulates the PI3K pathway, which is important in establishing its central role in multiple cancers [20]. These examples of interaction in multiple human cancers identify FLNA as an important regulator of tumor cell signaling and growth, and thereby make FLNA attractive in developing novel strategies to stop or at least to slow the growth of human tumors.

## 3. Approaches Targeting Cleavage of Filamin A in Cancers

FLNA is considered to have an important function in cancer development and the progression to metastasis. By being present in the cytoplasm and interacting with oncogenic cell signaling molecules, FLNA can promote cancer cell growth and metastasis as well as when present in the nucleus as a part of the transcriptional complex.

FLNA 280 kDa consists of 24 immunoglobulin-like repeat modules interrupted by two hinge regions between repeats 15–16 and 23–24 [2] (Table 2). These hinge domains of FLNA are proteolyzed by cysteine protease calpain, producing a 90-kDa carboxyl-terminal fragment (FLNA^CT^). Calpain is a calcium-activated protease which exists as an inactive proenzyme in the cytosol. When intracellular calcium levels are overloaded, it triggers a conversion of the proenzyme to its active form and is expressed ubiquitously in mammals [21]. Calpains are a conserved family of calcium-dependent cysteine proteinases with ubiquitous or tissue-specific expression. The Calpain family consists of 15 isoforms, in which calpain 1 (µ-type) preferentially digests FLNA as its main substrate. Although the physiological role of calpains is still incompletely understood, they actively participate in multiple cellular processes. Human mutations in the calpain 1 gene (*CAPN1*) have been associated with cerebellar ataxia, hereditary spastic paraplegia, and spinal muscular atrophy [22]. In addition to genetic disorders, different cancer forms are being linked to tissue-specific calpains. Interestingly, an increased expression of calpain small subunit 1 in multiple forms of human cancer is associated with shorter survival [23]. In addition, calpain small unit 1 has a central role in the metastasis of human cancers such as nasopharyngeal [24] and gastric [25] cancer. Whether calpains exert these effects via the FLNA substrate remains to be explored. In addition to Ca^++^-dependent activation, a hypoxic stimulus induces calpain activity, which increases cleavage of FLNA. This mechanism producing FLNA^CT^ is a critical regulator of VEGF-mediated angiogenesis in growing tumors [12]. To block calpain activity, calpeptin, a potent cell-permeable inhibitor, has been developed and tested in vitro to inhibit tumor cell growth [26,27]. Thus, the exciting data on the mechanisms and underlying signaling pathways of calpains in cancer growth and metastasis make use of calpain inhibitors attractive for aggressive tumor therapies. However, one potential side effect of calpain inhibition is impairment in wound healing after surgery [28]. FLNA is involved in human intrahepatic cholangiocarcinoma progression, in which calpeptin strongly decreases FLNA^CT^ expression, reducing cell proliferation and migration [29].

Tripartite motif containing 44 (TRIM44) regulates the DNA damage response in cancer cells with intact autophagy by preventing degradation of FLNA, increasing DNA damage repair [30]. Thus, TRIM44 serves as a potential therapeutic target for therapy-resistant tumor cells. In addition, TRIM44 promotes BRCA1 functions in homologous recombination repair to induce cisplatin chemoresistance in lung adenocarcinoma patients by deubiquitinating FLNA [31]. This provides potential therapeutic targets with cisplatin resistance. FLNA is also ubiquitinylated by ASB2α, an E3 ubiquitin ligase specificity subunit, regulating cell spreading and triggering proteasomal degradation of FLNA by targeting the calponin homology 1 domain [32].

Megakaryoblastic leukemia 1 (MKL1) is a coactivator of serum response factor (SRF) that promotes the expression of genes associated with cell proliferation, motility, adhesion, and differentiation processes that also involve dynamic cytoskeletal changes in the cell [33]. In this regard, FLNA functions as a positive cellular transducer linking actin polymerization to MKL1-SRF activity, counteracting the known repressive complex of MKL1 and monomeric G-actin. Thus, the correlation of FLNA expression with cancer stages and patient prognosis, as well as the clarification of its nuclear or cytoplasmic localization and its dual role (promoting or suppressing) in different cancers, can be observed [34].

Klotho protein decreases melanoma cell invasive potential by negatively regulating Wnt5A-mediated FLNA cleavage [35]. This mechanism is involved in the inhibition of calpain, resulting in decreased FLNA cleavage, which is critical for melanoma cell motility and invasive potential. In these melanoma cells, the use of 1,2- Bis(2-aminophonoxy)ethane-N,N,N′,N′-tetraacetic acid tetrakis-(acetoxymethylester) (BAPTA-AM) as a calcium chelator also decreases FLNA cleavage, implying that both ROR2 and calcium are required for Wnt5A-mediated FLNA cleavage, potentially through the activation of calpains [36]. In highly metastatic prostate cancer cells, extracellular Ca^++^ induces cleavage of FLNA. Treatment of these cancer cells with leupeptin, a protease inhibitor, and/or ALLM, a calpain specific inhibitor, blocks extracellular Ca^++^-mediated cleavage of FLNA and thereby reduces cellular migration in androgen receptor deficiency [37].

Another enzyme that cleaves FLNA is the granzyme B, as its cleavage accounts at least partly for caspase-independent cell death mediated by the proapoptotic protease granzyme, releasing ~110 and 95 kDa fragments, as observed in Jurkat cells [38]. In addition, FLNA is cleaved by the protease caspase-3, releasing 170, 150, and 120 kDa major FLNA^NT^, and 135, 120, and 110 kDa major FLNA^CT^ from 280 kDa when apoptosis is induced by etoposide in both the human monoblastic leukemia cell line U937 and the human T lymphoblastic cell line Jurkat [39]. The cleavage of FLNA is blocked by a cell permeable inhibitor of caspase-3-like protease, Ac-DEVD-cho, but not by an inhibitor of caspase-1-like protease, Ac-YVAD-cho.

In experimental models, plasmid vectors expressing H1-deleted form of FLN have been developed to study cellular responses in presence of uncleavable form of FLNA [12]. Melanoma cells expressing FLNA without H1 domain display impaired migration. In the presence of this uncleavable form of FLNA, nuclear shuttling signaling, and transactivation domain activity of HIF-1a is reduced [12]. Consequently, induction and secretion of VEGF is reduced, leading to impaired tumor angiogenesis and growth. In breast cancer cells, overexpression of mutant FLNA proteins resistant to calpain cleavage impairs focal adhesion disassembly at the leading edge of motile cells, in which an siRNA-resistant mutant FLNA has been designed by introducing a silent mutation at position 3330 (gac→gat) using a standard site-specific mutagenesis assay [40]. In prostate cancer cells, expression of a calpain-resistant FLNA mutant (Δ1.762–1.764) inhibits the increase in transactivation activity observed for transcriptional coactivator FHL2 following ionomycin-stimulated calpain activation [41].

**Table 2 cells-13-01394-t002:** Strategies targeting inhibition of FLNA cleavage and cellular outcomes in cancer cells.

Cancer Type	Cell Lines	Inhibition of FLNA Cleavage	Outcome of FLNA^CT^ Inhibition	Reference
Glioblastoma multiforme	U87 A172	Calpeptin	Increased cell growth and invasion	[27]
Melanoma, Prostate Fibrosarcoma	A7 PC3 T241	Calpeptin	Reduced cell proliferation, migration, and colony formation	[26]
Melanoma	A7 M2	Calpeptin Mutant protein resistant to calpain cleavage	Reduced tumor angiogenesis by HIF-1α/VEGF signaling	[12]
Breast	MCF-7 MDA-231 BT-20	Mutant protein resistant to calpain cleavage	Increased cell migration and invasion by focal adhesion disassembly	[40]
Prostate	COS1 LNCaP DU145 22Rv1	Calpeptin	Decreased FHL2 accumulation and decreased AR co-activation	[41]
Prostate	DU145 PC-3	Leupeptin and ALLM	Reduced cell migration	[37]
Melanoma	UACC903 M93-047	Klotho protein	Inhibition of cell motility by Wnt5A signaling	[35]
Melanoma	UACC90 M93-047 UACC647 Franklin Square G-361	BAPTA-AM chelation of intracellular calcium	Inhibition of cell motility by Wnt5A signaling	[36]
Melanoma	M2 A7	Granzyme B	Inhibition of caspase-independent cell death	[38]
Leukemia and lymphoma	U937 Jurkat	Caspase 3 inhibitor Ac-DEVD-cho	-	[39]

## 4. Future Directions

The current challenge is to determine which role interaction FLNA plays in the organization of the actin cytoskeleton, cellular signaling and transcription, and to identify the biochemical mechanisms that regulate these interactions for developing inhibitory strategies in particular types of human cancers. Many conclusions on FLNA function have been provided based on cell culture models, although there are only few studies in in vivo tumors, making future studies challenging. Both increased and decreased levels of FLNA expression correlate with tumor grade and patient survival depending upon cancer type or stage. Novel discoveries of FLNA functions in cancer development are still underway, but already FLNA is considered as an important player in cancer development and the progression to metastasis. Development of blocking strategies to inhibit FLNA cleavage and subsequent localization to the nucleus could be a new and potent field of research in treating cancer. As calpain contributes to tumor cell migration, invasion, and metastasis by altering focal adhesion dynamics and promoting cytoskeletal remodeling, inhibitors of calpain have been developed and tested in experimental models of cancers despite considerable debate about the specificity. These peptides are generally questioned for cellular permeability and pharmacokinetics; however, the blood–brain barrier permeability of these peptidomimetic inhibitors have been recently reported [42]. Nevertheless, agents to inhibit calpains in humans have yet to be approved.

Protein mimotopes, or blocking peptides, are small therapeutic peptides that may be designed to prevent FLNA^CT^ from interacting with its critical partners by selectively targeting specific binding sites (Figure 3 and Figure 4). Challenges in blocking specific peptide designs, production in sufficient quantities, high binding affinity, possible off-target effects, half-life, and mode of administration remain for the potential use of these therapeutic peptides [43]. To identify novel strategies in blocking specific binding sites of FLNA partners, high-throughput screens of small molecules are also promising. Such a small molecule, kartogenin, binds to FLNA, disrupting its interaction with the transcription factor core-binding factor β subunit (CBFβ), and induces chondrogenesis by regulating the CBFβ-RUNX1 transcriptional program in osteoarthritis [44]. Another rapidly growing exciting front is the development of nanobodies in a wide variety of research fields, particularly in the diagnosis and treatment of diseases [45]. To increase affinity, stability and expression levels, libraries of small single domain antibodies containing antigen binding variable domains of the heavy chain are being screened in exciting disease models as they easily penetrate tissues, are humanized for pharmaceutical applications, and are converted into different formats such as Fc-fusion proteins and heterodimers as modular building blocks targeting intracellular proteins. Prediction and analysis results provide helpful information to reveal the regulatory mechanism of calpain cleavage in different cancer types. The design and development of such nanobodies which bind to the H1 domain of FLNA to occupy motifs that are used for proteolysis by calpain (Figure 3), specific sites of FLNA^CT^ interacting with transcriptional factors, or cellular signaling molecules that are critical for carcinogenesis are attractive approaches. Membrane penetration capabilities, the ability to recognize unique antigens, and tumor cell specificity are the future challenges of cancer therapeutics. To this end, such antibodies must be specifically delivered inside the living tumor cell. Nevertheless, targeting B-cell maturation antigens, nanobodies have recently been approved to treat human refractory/relapsed multiple myeloma [46]. For nanobody-mediated targeting strategy into the cells, viral vectors such as adenovirus-associated viral vectors have been utilized locally or systemically in preclinical models of human cancer [47].

In cancer chemotherapy, a combination of new technologies will allow the rapid advancement of efficient therapeutic treatments. Thus, using these potentially exciting technologies to block cleavage of FLNA in cancers would stop subsequent nuclear translocation, impair signaling of various transcription factors, and reduce tumor cell migration and growth.

## Figures and Tables

**Figure 1 cells-13-01394-f001:**
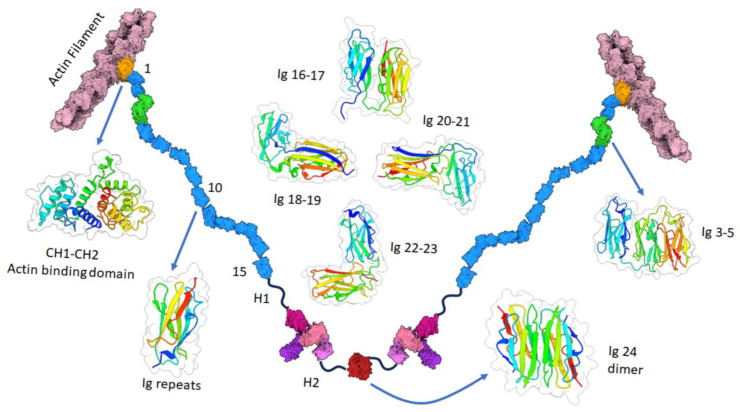
Actin bound filamin A structure. The model was built by using alpha fold filamin A model, crystal structures, and a cryo-EM structure. The PDB ID of the experimental models used in the figure are 6D8C (CH1 and F-actin complex), 3HOP (CH1-CH2), 4M9P (Immunoglobulin (Ig) repeats 3–5), 2K7P (Ig16-17), 2K7Q (Ig18-19), 2J3S (Ig19–21), 4P3W (Ig20-21), and 3CNK (Ig24 dimer).

**Figure 2 cells-13-01394-f002:**
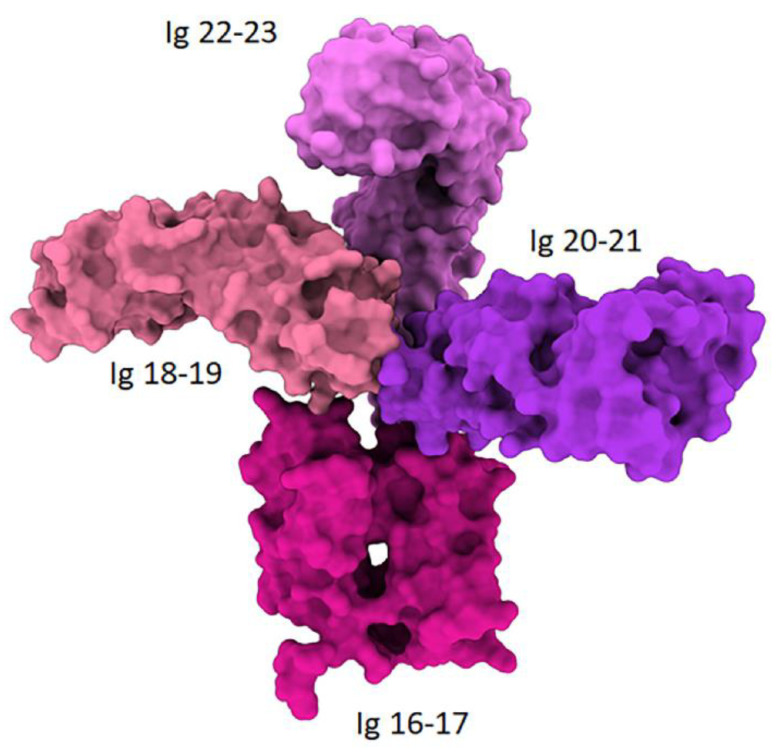
Theoretical organization of Ig repeats between H1 and H2 region. The experimental 3D structures suggest a non-linear organization of Ig repeats at this region [2K7P (Ig16-17), 2K7Q (Ig18-19), 2J3S (Ig19–21), 4P3W (Ig20-21)].

**Figure 3 cells-13-01394-f003:**
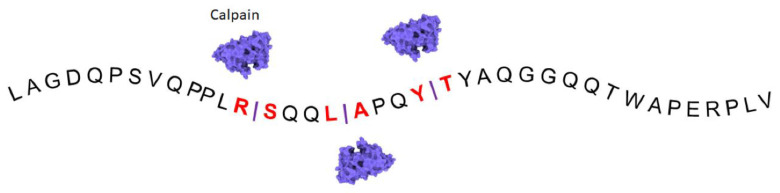
Calpain cleavage sites in the H1 region. Calpain cleavage sites are predicted using the online predictor tool (DeepCalpain) in the Hinge 1 region containing 38 amino acids.

**Figure 4 cells-13-01394-f004:**
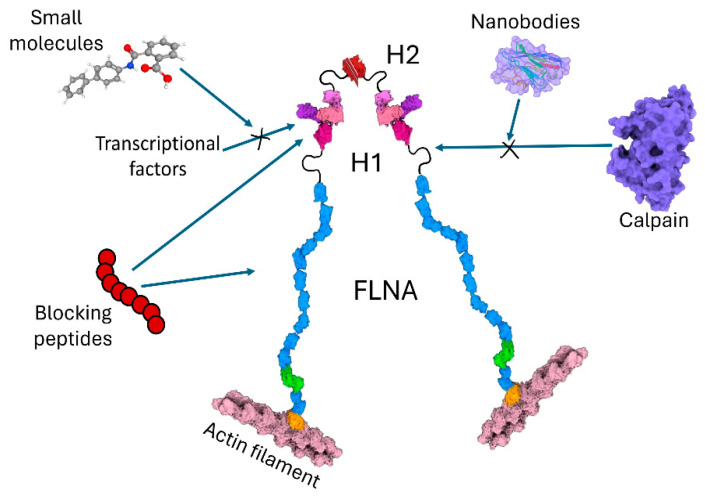
Future directions for targeting to block either cleavage of FLN^CT^ or interaction of FLNA^CT^. Small molecules that bind to partners of FLNA, nanobodies interacting with calpains, or peptides which block specific IgG domains of FLNA will be designed and utilized in future strategies to treat human cancers.

**Table 1 cells-13-01394-t001:** Cellular outcomes mediated by interaction of FLNA^CT^ in cancer cells.

Interaction Partner	Cellular Outcome	Cancer Types	Cell Lines	Reference
HIF-1α	Nuclear localization, promotes tumor growth and angiogenesis	Melanoma	M2 A7	[12]
STAT3	Increases MYCN expression and aggressiveness of tumors	Neuroblastoma	SKNBE2 SHSY5 Kelly IMR-32	[13]
SMAD2	Increases C-MET expression and tumor cell migration	Liver Pancreas Prostate Lung	HepG2 H69 BON PC-3 M2 A7	[14]
Androgen receptor	Nuclear localization, promotes tumor growth	Prostate	LNCaP PC3 DU145 MCF-7 T47D	[15]
BRCA1	Stabilizes DNA repair, acts as checkpoint in tumor progression	Breast	HCT116 293FT M2 A7	[16]
BRCA2	DNA damage response pathways in tumors	Melanoma Breast Pancreas	M2 A7 MCF-7 Capan-1	[17]
GPCR	Down regulates PI3K pathway in tumor signaling	Melanoma Neuroblastoma Pancreas	BON CHO M2 A7 HaCaT SH-SY5Y BxPC3	[20]

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
