# Peer review of "Targeting Cleavage of C-Terminal Fragment of Cytoskeletal Filamin A in Cancers"

_cells, 2024, doi:10.3390/cells13161394_

Round 1

Reviewer 1 Report

Comments and Suggestions for Authors

The perspective article by Cakici et al. is an interesting review about the function of a proteolytic fragment of filamin A , termed here FLNACT , and how it may be involved in cancer phenotypes. Many of the functions of FLNACT are nuclear, and it has been shown that it localizes in nucleus and is involved in nuclear trafficking, transcription regulation and DNA repair, for instance. However, FLNA mainly functions in cytoplasm and in many cases, it is not clear if the cancer-related phenotypes are caused by nuclear function of FLNACT   or cytoplasmic functions of FLNa.

The challenge in many of the studies is that they are based on cell culture models and there are only a couple studies of FLNA function  in in vivo tumors (McGrath, 2013, Vitali 2024).  Many of the tumor studies show correlation between FLNACT expression and tumor grade of patient survival. In some cases, higher grade tumors have decreased FLNACT levels, in other cases elevated. This may depend on cancer type or on other. Like vice, in some cell lines elevation of FLNACT  level increases cell migration, in others decreases.

I think this article is well written and should be published. I suggest that the authors would emphasize a bit more the varying outcomes of FLNACT  manipulations depending on cancer types or cell models (see comments 11 and 12) and in addition I present below some suggestions for small corrections or clarification for the text.

My detailed comments and suggestions to the Authors.

1 line 44

Because nuclear localization of FLNACT is the main focus of the review, I would like the authors to cite the original article demonstrating the finding (with perhaps some recent confirmation), instead of their own review from 2010 (ref 3)

2. Line 82

“In melanomas, transcriptional factor HIF-1a interacts with FLNACT ….”

Since many of the studies have been conducted using tumor derived cells lines instead of primary tumors, I ask the authors to be more specific about the models used. Here I would use …melanoma cells… . This said, I congratulate the authors for providing the details of the cell lines used in Table 1.

3. line 83

“… increase tumor progression..”  : could this be said more specifically … xenograft tumor progression..

4. line 84

“In high-risk neuroblastoma….” : This refers to the authors’ own work. I ask the authors to specify here what was studied in high-risk neuroblastoma tumors and what was studied in cell lines. My understanding is following.  FLNA levels are elevated in high-risk neuroblastomas, FLNA shRNA reduces neuroblastoma cell xenograft growth. FLNA interacts with STAT3  and FLNA shRNA reduced STAT3 phosphorylation and transcriptional activity in neuroblastoma cells.

5. lines 85-86

“In tumors that are induced by HGF/c-Met….” (this is authors’ own work): could this also be …tumor cells….  . The authors show FLNA expression in adenocarcinomas, the rest of the work is done in various tumor cell models.

6. lines 86-89

“Particularly in prostate cancer cells…. “. I ask the authors to check this sentence. My understanding from ref 14 (Di Donato et al 2021) is that they showed AR expression is cancer associated fibroblasts (CAFs)  and that inhibition of AR-FLNA interaction in CAFs reduces the invasiveness of prostate cancer cells in a co-culture assay.

7. line 89

“In breast cancer, ….” Could this be …breast cancer cells…

8. lines 96 and 130

This is a general comment to the referencing system in Table  1 and Table 2

It would be helpful to the reader if same referencing system would be used in the Tables as elsewhere in the manuscript. Or, alternative, the numeric reference could be given in addition to the first author and year.

9. Line 108

Filamin degradation by Calpains. Please, check the reference. In ref 17 Nakamura et al. studied calmodulin binding to FLNA ABD.

10. line 123

Testing calpeptin in vivo. I was not able to find results of using calpeptin in xenograft model in ref 11. Please, check if this reference is correct.

11. Line 130+. Table 2 Cai et al 2020

According to my understanding ref 23 (Cai et al., 2020) shows that calpain suppresses and calpeptin induces glioblastoma cell migration and invasion. Thus, the outcome of FLNACT inhibition is wrong in the table. This is an example of opposite effects of FLNCT in cell migration depending on cancer cell types.

12. line 130+, Table 2 Xu et al., 2010

The table now claims that outcome of FLNACT inhibition was suppression of cell migration and invasion by focal adhesion assembly (ref 35, Xu et al., 2010). I understood that in this paper FLNA siRNA reduced cells migration that was rescued by overexpression of FLNa, but not FLNA that cannot be calpain cleave. Thus, outcome of FLNACT inhibition was induced cell migration.

13. line 130+, Table 2 McGrath et al, 2013

Is this an error in table, please check? In ref 35 (McGrath et al, 2013) FLNACT stimulation

increased nuclear  FHL2  and androgen response.  Outcome of calpain

inhibition and calpain resistant FLNA was decreased FHL2 accumulation and decrease of AR co-activation. Calpeptin could be mentioned as an inhibition method.

14. Line 131

“Tripartite motif containing 44… .“ Please, rephrase this sentence. Now this is almost direct copy from the original reference (26) and rather complicated. You could do the readers a favor in explaining this in more common language and perhaps also mention the connection to BRCA1 mentioned earlier, if there is any. I have to admit that I did not fully understand the role of filamin here. Was the role cytoplasmic or nuclear? There was a mention that TRIMM prevents proteosome degradation of FLNA, resulting to increase of FLNA in the nucleus whereas TRIM44 silencing decreased nuclear FLNA.

15. Line 131 onwards

Comment related to ubiquitinylation. Since TRIM44 is a deubiquitinase, and the section is under the heading: Approaches targeting FLNA in cancers, it might be appropriate to take this out as a separate paragraph and also mention that FLNA is ubiquitinylated by ASB2, which also has other targets. Discussion of  MKL-1 could then be separated from this part.

16. Line 207-208

“Prediction and analysis results provide helpful information to reveal the regulatory mechanism of calpain cleavage in different cancer types  (26)”

I was puzzled about this sentence and the reference. Ref 26 did not mention calpain or nanobodies in any way . This sentence does not seem to link to the text before and after. Could this be removed?

Author Response

Response to the Comments raised by the Reviewer 1

FLNA mainly functions in cytoplasm and in many cases, it is not clear if the cancer-related phenotypes are caused by nuclear function of FLNACT   or cytoplasmic functions of FLNA.

We consider that either cytoplasmic or nuclear FLNA or both are involved in cancer-related phenotypes depending upon type and stage of cancer, where cytoplasmic FLNA is particularly involved in cancer cell migration and proliferation, while nuclear FLNA in cancer cell signaling and transcription. This sentence has now been included.

The challenge in many of the studies is that they are based on cell culture models and there are only a couple studies of FLNA function in in vivo tumors (McGrath, 2013, Vitali 2024).  Many of the tumor studies show correlation between FLNACT expression and tumor grade of patient survival. In some cases, higher grade tumors have decreased FLNACT levels, in other cases elevated. This may depend on cancer type or on other. Like vice, in some cell lines elevation of FLNACT  level increases cell migration, in others decreases.

 We thank for this comments which has now been included.

1 line 44. Because nuclear localization of FLNACT is the main focus of the review, I would like the authors to cite the original article demonstrating the finding (with perhaps some recent confirmation), instead of their own review from 2010 (ref 3)

Reference 3 has been given together with our recent review instead of adding numerous references after that sentence.

  1. Line 82. “In melanomas, transcriptional factor HIF-1a interacts with FLNACT ….” Since many of the studies have been conducted using tumor derived cells lines instead of primary tumors, I ask the authors to be more specific about the models used. Here I would use …melanoma cells…. This said, I congratulate the authors for providing the details of the cell lines used in Table 1.

As suggested, “In melanomas” has been replaced with “melanoma cell lines”.

  1. line 83. “… increase tumor progression...”: Could this be said more specifically … xenograft tumor progression..

 As suggested, “…increase tumor progression…” has been replaced with “…increase xenograft tumor progression…”.

  1. line 84. “In high-risk neuroblastoma….”: This refers to the authors’ own work. I ask the authors to specify here what was studied in high-risk neuroblastoma tumors and what was studied in cell lines. My understanding is following. FLNA levels are elevated in high-risk neuroblastomas, FLNA shRNA reduces neuroblastoma cell xenograft growth. FLNA interacts with STAT3 and FLNA shRNA reduced STAT3 phosphorylation and transcriptional activity in neuroblastoma cells.

As pointed, the clinical or experimental sources of data have been clearly mentioned.

  1. lines 85-86. “In tumors that are induced by HGF/c-Met….” (this is authors’ own work): could this also be …tumor cells…. . The authors show FLNA expression in adenocarcinomas, the rest of the work is done in various tumor cell models.

 As suggested, “tumors” has been replaced with “multiple tumor cell lines”.

  1. lines 86-89. “Particularly in prostate cancer cells…. “. I ask the authors to check this sentence. My understanding from ref 14 (Di Donato et al 2021) is that they showed AR expression is cancer associated fibroblasts (CAFs) and that inhibition of AR-FLNA interaction in CAFs reduces the invasiveness of prostate cancer cells in a co-culture assay.

As suggested, “prostate cancer cells” has been replaced with “cancer associated fibroblasts”.

  1. line 89. “In breast cancer, ….” Could this be …breast cancer cells…

As suggested, “breast cancer” has been replaced with “breast cancer cells”.

  1. lines 96 and 130. This is a general comment to the referencing system in Table 1 and Table 2. It would be helpful to the reader if same referencing system would be used in the Tables as elsewhere in the manuscript. Or, alternative, the numeric reference could be given in addition to the first author and year.

As suggested, reference numbers are provided in both Table 1 and 2.

  1. Line 108. Filamin degradation by Calpains. Please, check the reference. In ref 17 Nakamura et al. studied calmodulin binding to FLNA ABD.

We thank for this comment and provide the correct reference now.

  1. line 123. Testing calpeptin in vivo. I was not able to find results of using calpeptin in xenograft model in ref 11. Please, check if this reference is correct.

Thank you for pointing this incorrect information. We have deleted this reference from the end of that sentence.

  1. Line 130+. Table 2 Cai et al 2020. According to my understanding ref 23 (Cai et al., 2020) shows that calpain suppresses and calpeptin induces glioblastoma cell migration and invasion. Thus, the outcome of FLNACT inhibition is wrong in the table. This is an example of opposite effects of FLNCT in cell migration depending on cancer cell types.

Thank you for pointing this incorrect information. We have now re-written it as “Increased” in Table 2.

  1. line 130+, Table 2 Xu et al., 2010. The table now claims that outcome of FLNACT inhibition was suppression of cell migration and invasion by focal adhesion assembly (ref 35, Xu et al., 2010). I understood that in this paper FLNA siRNA reduced cells migration that was rescued by overexpression of FLNa, but not FLNA that cannot be calpain cleave. Thus, outcome of FLNACT inhibition was induced cell migration.

Thank you for pointing this incorrect information. We have now re-written it as “Increased” in Table 2.

  1. line 130+, Table 2 McGrath et al, 2013. Is this an error in table, please check? In ref 35 (McGrath et al, 2013) FLNACT stimulation increased nuclear FHL2 and androgen response. Outcome of calpain inhibition and calpain resistant FLNA was decreased FHL2 accumulation and decrease of AR co-activation. Calpeptin could be mentioned as an inhibition method.

Thank you for pointing this incorrect information. We have now re-written it as suggested “Decreased FHL2 accumulation and decreased AR co-activation” and changed it “Calpeptin” as mode of inhibition of FLNA cleavage in Table 2.

  1. Line 131. “Tripartite motif containing 44…“ Please, rephrase this sentence. Now this is almost direct copy from the original reference (26) and rather complicated. You could do the readers a favor in explaining this in more common language and perhaps also mention the connection to BRCA1 mentioned earlier, if there is any. I have to admit that I did not fully understand the role of filamin here. Was the role cytoplasmic or nuclear? There was a mention that TRIMM prevents proteosome degradation of FLNA, resulting to increase of FLNA in the nucleus whereas TRIM44 silencing decreased nuclear FLNA.

We have now rephrased this sentence in more common language and mentioned information on the connection to BRCA1.

Lin Lyu et al. proposed that TRIM44 increases nuclear FLNA expression and downregulation of p62 stabilizes nuclear FLNA. However, they did not study the role of FLNA cleavage or inhibition of FLNA cleavage by calpain inhibitor calpeptin.

  1. Line 131 onwards. Comment related to ubiquitinylation. Since TRIM44 is a deubiquitinase, and the section is under the heading: Approaches targeting FLNA in cancers, it might be appropriate to take this out as a separate paragraph and also mention that FLNA is ubiquitinylated by ASB2, which also has other targets. Discussion of MKL-1 could then be separated from this part.

Information on TRIM44 has been separated from discussion of MKL-1 as a new paragraph. In this paragraph, ubiquitination of FLNA by ASB2 has also been mentioned as suggested.

  1. Line 207-208. “Prediction and analysis results provide helpful information to reveal the regulatory mechanism of calpain cleavage in different cancer types (26)”. I was puzzled about this sentence and the reference. Ref 26 did not mention calpain or nanobodies in any way. This sentence does not seem to link to the text before and after. Could this be removed?

We thank for this incorrect citation and thus removed the reference from this sentence.

Reviewer 2 Report

Comments and Suggestions for Authors

 This review summarized some of the recent studies validating filamin-A, particularly the cleavage of filamin-A, as a target for cancer treatment. It is of value to cancer biologists and investigators who are interested in filamin-A. The review was largely well-structured.

Table 1 provided a selected list of interaction proteins relevant to cancer. The important interaction of filamin-A with BRCA2, and filamin-A’s roles in G2/M transitions and sensitivities to chemotherapy agents were not listed in this table. 

Comments on the Quality of English Language

N/A

Author Response

Response to the Comments raised by the Reviewer 2

Table 1 provided a selected list of interaction proteins relevant to cancer. The important interaction of filamin-A with BRCA2, and filamin-A’s roles in G2/M transitions and sensitivities to chemotherapy agents were not listed in this table.

Interaction of BRCA2 with FLNACT has now been included in Table 1 as suggested and proposed mechanisms have been accordingly mentioned in the section “Interaction partners of filamin A in cancers”.